# A Hybrid Deep Learning Model for Brain Tumour Classification

**DOI:** 10.3390/e24060799

**Published:** 2022-06-08

**Authors:** Mohammed Rasool, Nor Azman Ismail, Wadii Boulila, Adel Ammar, Hussein Samma, Wael M. S. Yafooz, Abdel-Hamid M. Emara

**Affiliations:** 1School of Computing, Faculty of Engineering, Universiti Teknologi Malaysia, Skudai, Johor Bahru 81310, Malaysia; azman@utm.my (N.A.I.); hussein.samma@utm.my (H.S.); 2Robotics and Internet-of-Things Laboratory, Prince Sultan University, Riyadh 12435, Saudi Arabia; wboulila@psu.edu.sa (W.B.); aammar@psu.edu.sa (A.A.); 3Department of Computer Science, College of Computer Science and Engineering, Taibah University, Medina 42353, Saudi Arabia; aemara@taibahu.edu.sa; 4Department of Computers and Systems Engineering, Faculty of Engineering, Al-Azhar University, Cairo 11884, Egypt

**Keywords:** brain tumour, MRI images, deep learning, CNN, Google-Net, SVM, fine-tuning

## Abstract

A brain tumour is one of the major reasons for death in humans, and it is the tenth most common type of tumour that affects people of all ages. However, if detected early, it is one of the most treatable types of tumours. Brain tumours are classified using biopsy, which is not usually performed before definitive brain surgery. An image classification technique for tumour diseases is important for accelerating the treatment process and avoiding surgery and errors from manual diagnosis by radiologists. The advancement of technology and machine learning (ML) can assist radiologists in tumour diagnostics using magnetic resonance imaging (MRI) images without invasive procedures. This work introduced a new hybrid CNN-based architecture to classify three brain tumour types through MRI images. The method suggested in this paper uses hybrid deep learning classification based on CNN with two methods. The first method combines a pre-trained Google-Net model of the CNN algorithm for feature extraction with SVM for pattern classification. The second method integrates a finely tuned Google-Net with a soft-max classifier. The proposed approach was evaluated using MRI brain images that contain a total of 1426 glioma images, 708 meningioma images, 930 pituitary tumour images, and 396 normal brain images. The reported results showed that an accuracy of 93.1% was achieved from the finely tuned Google-Net model. However, the synergy of Google-Net as a feature extractor with an SVM classifier improved recognition accuracy to 98.1%.

## 1. Introduction

In recent years there has been an apparent increase in interest in brain tumour diseases, which affect humans severely and are life threatening. Brain cancer is the tenth most-common primary reason for death in men and women. As per the International Agency for Research on Cancer, approximately 126,000 people worldwide are diagnosed with brain tumours each year, with a death rate of over 97,000 [1]. Survival rates for people with cancerous brain tumours, on the other hand, vary greatly and are determined depending on several factors, including the patient’s age and the type of brain tumour. Brain tissue is complex; the normal tissues consist of three types of main tissues known as white matter (WM), gray matter (GM), and cerebral spinal fluid (CSF).

While abnormal tissues such as tumours, necrosis, and oedema. Necrosis is the death of cells inside an active tumour, whereas oedema occurs near the active tumour boundaries [2,3,4]. Malignant destructive tumours grow quickly and spread to other tissues, whereas benign tumours grow slowly and do not spread or invade other tissues [5,6]. Brain tumours are classified into three types based on these two main categories: gliomas, meningiomas, and pituitary tumours. Glioma tumours form in brain tissues other than nerve cells and blood vessels. In contrast, meningioma tumours grow in the membrane surface that covers the brain and surrounds the central nervous system, and pituitary tumours form inside the skull [7,8]. The World Health Organization has classified brain tumours into several types. This classification is based on the origin of the cell and the cell’s behaviour, which ranges from less aggressive to more aggressive [7,9,10]. The most significant difference features between these three types of tumours are that meningiomas are typically benign and slow growing, whereas gliomas are typically malignant, in contrast to pituitary tumours, which, even if mild, can cause other medical problems [7,8]. Due to the preceding information, identifying these three different types of tumours is an essential step in the clinical diagnosis process of patients.

Medical experts have claimed for many years that detecting brain tumours in clinics with human interpretation is difficult. Due to this, there is an urgent need for more reliable early tumour detection techniques such as computer-aided diagnosis (CAD) [10,11,12]. CAD techniques have been a crucial solution in various medical applications that rely on feature extraction from medical images, such as trying to distinguish between healthy and abnormal tissue [13,14,15].

Medical imaging techniques are critical in detecting tumours early and improving treatment options. Brain tumours are studied using various non-invasive imaging techniques, including CT, MRI, SPECT, PET, and X-ray [16,17]. The use of non-invasive technologies to detect brain tumours is an important step in the treatment process [18]. Medical imaging techniques can provide information such as the position, volume, shape, and category of brain tumours to aid in diagnosis. MRI is regarded as a typical technique for providing detailed information on the anatomical tissue of humans due to its widespread ability to capture the definition of soft tissue compared with other medical image techniques [17,19].

Image classification is a machine learning (ML) method used to train a computer system to be an “expert system”. It uses classified medical images of the medicine domain for diagnostic and teaching purposes. The classification process of medical images is based on pre-processing by extracting features; for example, brain images are used to determine and classify the type of tumour [20,21]. Numerous classification techniques, including PNN, K-NN, ANN, SVM, and BPNN, can be used for classification and applied to a wide range of data sets, including medical image datasets [22,23]. 

Deep learning (DL) is the current state of the art, having gained popularity and widespread interest in every field, particularly medical image analysis [24]. Deep learning achieves greater power and flexibility due to its ability to process a large amount of unstructured data by passing it through several layers; each layer can extract features incrementally and pass them to the next layer [25,26,27,28,29]. The main characteristic of the deep learning technique is that it focuses on automatically extracting features that represent data representations. Convolutional neural networks (CNN) are the most commonly used deep learning technique in the medical image domain. The CNN model achieves efficient processing capability for automatically extracting structured data features and representations from medical images [30,31].

However, this study uses a new brain tumour classification method based on the hybrid CNN-SVM method through MRI brain images. The contribution of this study could be summarized in the following points: It introduces a hybrid deep learning model that detects brain tumours in the early stages in order to accelerate the treatment process and control the spread of the malignant tissues.It proves that using hybrid deep learning classification, which combines Google-Net with SVM, gives higher accuracy and better results than traditional methods.It helps radiologists avoid errors from manual diagnosis of tumours through magnetic resonance images (MRI) images without invasive measures.

The remainder of this paper is structured as follows. Section 2 goes over related work. Section 3 explains the method and specifics of the proposed approach. Section 4 explains the outcome and discussion of the proposed approach. Section 5 contains the calculation. Table 1 contains a list of all the abbreviations used at work.

## 2. Related Work

ML and DL methods have recently been widely used for the detecting and classification brain tumours using various imaging modalities, particularly those obtained using MRI brain images. This section presents the most relevant and recent research work related to the study in this paper. Bahadur et al., 2017 [32] proposed an algorithm that uses MRI images to extract brain tumour segmentation, feature extraction, and classification data. To eliminate the effect of extraneous noise, they began by employing the signal-to-noise ratio method. To remove texture characteristics, the GLCM technique was used. Finally, feature vectors and tumour location data were used to segment the images by the BWT technique and identify the tumour type by using the SVM classifier. Amin et al., 2017 [33] proposed an automated system of classifying brain tumours using MR images with three major steps. First, various methods, such as strength, are used to segment the area of interest (ROI). Second, based on form, texture, and strength, the best features for each applicant lesion are selected. Finally, an SVM classifier is used to differentiate between cancerous and non-cancerous images. H. Ayad et al., 2018 [34], presented an approach for classifying MRI brain images as having normal tissue or abnormal tissue. At first, three function extraction techniques are used in this approach: GLCM, LBP, and HOG. A k-NN classifier is used to process the feature vectors obtained from each technique. A fusion operator is used to combine the classifiers’ dissimilarity test values to improve classification accuracy. In Emerson et al., 2018 [35], the authors proposed an approach divided into several steps for extracting features, and the local binary characteristics, grey-level co-occurrence features, and grey-level and wavelet features were extracted. These features were trained and classified using the SVM classifier M. J. Leo. In 2019, the authors in [36] developed a method for identifying brain tumours through MRI images. The first process in this study was to improve brain images using a median filter. The K-means clustering approach was then used to segment MRI brain tumour images. The GLCM approach extracted features for classifying MRI brain tumours, which would then be fed into the classifier for tumour classification using the k-NN technique. U.N. et al., 2020 [37] proposed a hybrid method for removing tumours through MRI images. In this process, the curvelet transformation was used to de-noise the image. Tumours were removed from brain images during the segmentation stage using ACO (ant colony optimization) and the Threshold process. Features extracted are classified into different categories based on shape and texture, and the features are prioritized before fusion using a concatenation-based method. Finally, the features that have been used are fed into the SVM classifier. In Kshirsagar et al. 2020 [38], the authors proposed a neural network approach to detect and classify brain tumours through MRI brain images. The pre-processing strategy used Gaussian filtering to remove noise, image enhancement, and image segmentation in the first stage. The GLCM technique was then used to extract features. For characterization, a neuro-fuzzy classifier is used. As part of the proposed work, the ANFIS technique was used. Kabir et al. 2020 [39] also proposed an approach for detecting brain tumours and extracting features through MRI images. An anisotropic diffusion filter was used with principal component-based grey-scale conversion to remove unwanted artifacts. Following that, the CLAHE technique was used to improve image contrast. The tumour was then segmented with multi-level thresholding and the Chan–Vese algorithm. Khawaldeh, Saed, et al. [40], based on a modified version of the Alex-Net model of CNN architecture, proposed a noninvasive grading system for glioma brain tumours. The classification was carried out using whole MRI brain images, with labels applied at the image level rather than the pixel level. Sajjad et al. [41] proposed a comprehensive data augmentation method for brain tumour classification using CNN. This method is used for the multi-grade classification of brain tumours and for segmenting the brain tumour through MRI brain images. They used transfer learning with a pre-trained VGG-19 model of CNN architecture to classify data. Fatih et al. [42] proposed a new approach that combines CNN architecture with neutrosophic expert maximum-fuzzy (NS-CNN) sure entropy for brain tumour type classification. The NS-CNN sure method for brain tumour segmentation was used in the segmentation process. Then, the images were fed into CNN architecture to extract tumour features and into the SVM technique to classify whether the tumour was benign or malignant. The authors proposed several classification methods based on ML and DL in their articles. They proposed a machine-learning and deep-learning algorithm for detecting and classifying two classes or three classes of brain tumours in MRI brain images. 

## 3. The Proposed Approach

This paper’s primary goal and motivation is to present a new hybrid deep-learning model method. The proposed approach investigated two different hybridizations. The first combines Google-Net (GN) with SVM and is named GN-SVM, and the second combines Google-Net with fine-tuning and is named GN-FT. Figure 1 illustrates the proposed methodology’s block diagram. The following subsections discuss the proposed approach in detail.

### 3.1. Fine-Tuned Deep Model

Fine-tuning is a process that takes a model that has already been trained (pre-trained) for one task and returns it or tweaks the same model to perform a classification task [43,44]. Assuming that the original and new tasks are similar, using an artificial neural network that has already been designed and trained allows us to leverage what the model has already learned rather than developing it from scratch [45,46]. 

The fine-tuning technique is utilized in this study by using a pre-trained neural network to transfer and classify MRI brain images. This method uses the Google-Net model combined with the fine-tuning technique (GN-FT) to generate an output. After feature extraction from MRI brain images using the convolution layers of the Google-Net model, the fine-tuning technique applies a full connection layer (FCL) to classify MRI brain images into four classes—glioma, meningioma, and pituitary tumours as abnormal tissue or non-tumours as normal tissue—based on extracted features.

The main steps to apply the fine-tuning technique on features extracted from the inputting images via a convolutional layer (CL) of the Google-Net model, as explained in Algorithm 1, are as follows. First, freeze the weights of the first few layers that have already from the already trained (pre-trained) network. This is followed by adding a new output layer (soft-max) into the target model, whose number of outputs is the number of categories in the target dataset. Finally, the Google-Net model will be ready to re-train the new dataset by setting the initial layers to zero. The freezing process of the layers increases the network’s training speed. Figure 2 shows the hybrid CNN architecture combined with the fine-tuning technique.
**Algorithm 1: Proposed Fine-Tuning Technique****Inputs: Training and testing images.****Outputs: Calculated accuracy**.Select the optimal value of features for determining Fine-Tuning output.Fine-Tuning Steps (condition of output limitation).**Step 1****:** *Load pre-trained* Google-Net model of CNN (replicates all model designs and their parameters on the Google-Net model, except the output layer)**Step 2:** *Truncate the pre-trained network’s last layer* (softmax layer) and replace it with our new output layer that is relevant to our problem.**Step 3:** *Add an output layer* to the target model, whose number of outputs is the number of categories in the target dataset.**Step 4:** *Freeze the weights of the pre-trained network’s first few layers.* The first few layers capture universal features such as curves and edges, which are also relevant to our new problem.**Step 5****:** *Start training* the new model structure while keeping those weights intact, with the network focusing on learning dataset-specific features in the subsequent layers.**Step 6:** The output Layer yields one of four classes: Not-Tumour.Glioma-Tumour.Meningioma-Tumour.Pituitary-Tumour.**Return accuracy**

### 3.2. Features Extraction

We used the finely tuned Google-Net from Section 3.1 for the feature extraction task. The feature extraction method is based on a pre-trained Google-Net. The Google-Net system operates as an arbitrary feature extractor, allowing the input image to propagate forward until it reaches the pre-specified layer (feature extraction layer). At this point, it stops and uses the outputs of that layer as our features. This proposed method uses the Google-Net model of pre-trained CNN deep learning. The Google-Net model is one of the CNN architectures proposed by Christian Szegedy of Google. The main processes of the Google-Net model integrate start-up layers with variable receptive fields provided by various kernel sizes [47]. These receptive fields generated operations in the new feature-map stack that captured sparse correlation patterns to extract features from images [48]. In this research, we used the Google-Net model to extract the features from MRI brain images in order to classify types of brain tumours.

The architecture of Google-Net is shown in Figure 3; it contains three various size filters (1 × 1, 3 × 3, and 5 × 5) for the same image and combines the features to get a robust output. It consists of 22 layers, and it lessens the number of parameters from 138 million to 4 million. The (1 × 1) convolution is introduced for dimension reduction. This architecture finds the best weight while training the network and naturally selects the appropriate features [49,50]. Figure 3 shows the dimensionality reduction process of convolution layers to feature extraction.

The depth of the Google-Net architecture is 22 layers with 27 pooling layers. There are nine inception modules in total, and they are stacked linearly. The endpoints of the inception modules are linked to the global average pooling layer. In this section, the finely tuned Google-Net was used to extract the MRI medical features from the last layer, which generated a feature vector of length 1024D, as shown in Figure 3.

### 3.3. Classification Subsection

The hybrid deep learning algorithm uses the finely tuned Google-Net model of the CNN technique based on two classification methods. First, it combines Google-Net with the support vector machine (SVM) classifier technique (GN-SVM). Second, it combines Google-Net with fine-tuning techniques (GN-FT), as shown in Figure 1. This hybrid deep learning algorithm is used to classify MRI brain images as normal tissue or abnormal tissue (tumour or not tumour) and classify the tumour types as glioma, meningioma, or pituitary tumours.

Support vector machines (SVM) are one of the common traditional machine learning classifiers and are related to supervised learning methods. The SVM technique is a commonly used classifier in various fields, including handwriting analysis, face analysis, medical image analysis, and so on. It is particularly useful for pattern detection and regression-based applications [51,52,53,54]. The SVM classification function works by separating the various groups and finding an optimal hyperplane to solve the learning problem. The SVM algorithm has three parts: simple concepts for linearly separable groups and expansion to the non-linearly separable case using kernel functions. One of the kernel functions is the radial basis function (RBF) kernel, often used when there is no prior knowledge of the results, and it generates a piecewise linear solution when discontinuities are suitable [55]. Figure 4 shows the process of the support vector machines (SVM) technique [55].

The SVM classifier technique is used in this proposed method to classify MRI brain images into four classes: glioma, meningioma, and pituitary tumours as abnormal tissue or non-tumours as normal tissue. After feature extraction from MRI brain images using convolution layers of the Google-Net model, the SVM algorithm will apply the full connection layer (FCL) as explained in Algorithm 2 to classify brain tumours based on the extracted features. Figure 5 describes the structure of the hybrid deep-learning classification model GN-SVM.
**Algorithm 2: The proposed SVM classifier technique****Inputs:** Training and testing images.**Outputs:** Calculated accuracy.Select the optimal value of cost and gamma for SVM.**While** (stopping condition is not met) **Do**
**Step 1:** Implement the SVM train step for each data point.**Step 2:** Implement SVM classification for testing data points.**Step 3:** Define SVM-based kernel as k(x,y)=exp exp (‖x−y‖σ).Where *x*, *y* belong to the samples of feature space in the training set parameter.**Step 4:** The objective is to have four classes:       1- Normal (Not Tumour)      2- Tumour, with three subclasses:1.**Glioma.**2.**Meningioma.**3.**Pituitary.****End While**
**Return** accuracy

## 4. Results and Discussion 

### 4.1. Dataset 

In this paper, we have applied data augmentation techniques to a large dataset of 3460 different types of brain MRI images [56]. The dataset was first posted online in 2017 by Jun Cheng and updated in 2020 by Sartaj Bhuvaji [57]. The image dataset includes 3064 T1-weighted contrast-enhanced MRI images from Kaggle.com. There are three main kinds of brain tumours: meningioma, which has 708 images; glioma, which has 1426 images; and pituitary tumours, which have 930 images. All pictures were collected from 233 patients in three planes: sagittal (1025 images), axial (994 images), and coronal (1045 images). The data was split randomly into training and testing groups, with 80% for training and 20% for testing. Each folder has four subfolders. These folders have MRIs of respective tumour classes [57]. Figure 6 illustrates the kinds of brain tumours, meningioma, glioma, pituitary, and not-tumour, from three planes: sagittal, axial, and coronal.

### 4.2. Evaluation Measures 

To evaluate the proposed use of hybrid deep-learning classification based on CNN architecture in the grading of brain tumours as glioma, meningioma, and pituitary tumours or as normal brain (No_Tumour), in this study, standard evaluation measures were used; the measures used in this study were precision, recall, and accuracy.

From the proposed method obtained by using the dataset, three statistical indices, namely true positive (*TP*), false positive (*FP*), false negative (*FN*), and true negative (*TN*), were calculated and used to evaluate the performance of the proposed classification system, as shown below [58,59].
(1)$$\mathrm{Accuracy}=\frac{TP+TN}{TP+FP+TN+FN}.$$
(2)$$\mathrm{Precision}\;=\frac{TP}{TP+FP}.$$
(3)$$\mathrm{Recall}\;=\frac{TP}{TP+TN}.$$

### 4.3. Performance Analysis 

All experiments were carried out and evaluations were conducted using MATLAB R2021b programming on a laptop computer equipped with an Intel Core-I5 processor, 20 GB of RAM, and a Hard SSD type. Full specification details are given in Table 2. This section presents the results of the proposed approach based on two combined methods—the first method, GN-SVM, and the second method, GN-FT—to identify tumour type through MRI brain images. Figure 7 and Figure 8 illustrate the overall confusion matrices. Based on these figures, we can conclude that the proposed system classified the brain tumours successfully and efficiently with accuracy values of 98.1% using the GN-SVM method and accuracy values of 93.1% using GN-FT for an input image size of 224 × 224.

As explained earlier, the dataset used in this study contains two types of MRI brain images, some having normal tissue and some having abnormal tissue. The normal images are considered as “not a tumour”, and the abnormal images as having different types of tumours, namely meningioma, glioma, and pituitary tumours. Table 3 tabulated the computed precision and recall measure for both methods (i.e., GN-SVM and GN-FT). As can be seen, GN-SVM produced better accuracy in terms of precision rate. This is due to the benefits of SVM in minimizing the false alarm rate. The average accuracy, precision, and recall rates for all types of tumours are shown in Figure 9.

### 4.4. Computational Analysis and Comparison with Other Algorithms 

This part of the research compares the execution time of two methods, GN-SVM and GN-FT. The execution time of the GN-SVM method is 0.097 s, and the GN-FT method is 0.098 s per single image, as Table 4 explains. As can be seen, both implemented approaches required almost the same classification time (i.e., testing time). 

Table 5 compares the accuracy of our proposed method with related work methods. As shown in the table, seven methods have been implemented for comparison purposes. It should be noted that we followed the same split of data, where 80% was used for training, and the remaining 20% was used for testing. As is indicated in Table 5, the best accuracy reached was 97.1 by the SVM + ROI + (RBF) + linear and cubic approach [28], which is a better accuracy than that achieved by our proposed method (CNN + Google-Net + fine-tuning). However, our second approach (Google-Net + SVM) seems to surpass all other methods with an accuracy of 98.1%. This is due to the power of finely tuned Google-Net in extracting useful features combined with the ability of SVM to perform well with high-dimensional classification problems.

### 4.5. Model Evaluation Using Public Dataset 

Further analysis was performed by validating the performance of the proposed approach with an additional benchmark dataset. We used the MRI brain images dataset, which was obtained from different patients gathered from several hospitals, WHO (World Health Organization), and the Whole Brain Atlas site, which was published by www.kaggle.com website: (https://www.kaggle.com/datasets/navoneel/brain-mri-images-for-brain-tumor-detection (accessed on 1 May 2022)). It contains 253 images of the normal brains of persons and 155 images of the brains of persons who have tumours, and 98 images of persons without tumour. Some sample images from the MRI brain dataset are shown in Figure 10. 

In this experiment, the dataset has been divided into 80% to 20% for training and testing, respectively. The outcomes of the proposed approach and other methods are given in Table 6. The results indicated that the proposed method is able to achieve the best results as compared with [34,36,40]. This is because the proposed hybrid model combines the benefits of the deep-learning model Google-Net in performing automatic feature extraction and the strength of SVM classifiers in performing classification. 

## 5. Conclusions

In this paper, we have presented a new hybrid deep-learning classification method based on CNN-SVM. This method combines a finely tuned Google-Net model with SVM to identify and classify the tumour types through MRI brain images and developed a new hybrid CNN architecture that efficiently automated the classification of MRI brain image datasets into four classes: meningioma, glioma, pituitary tumour, and not a tumour. The suggested technique outperforms existing deep-learning methods in recall, 98.1%, precision, 98.2%, and accuracy, 98.1%. The hybrid CNN method performed more rapidly among deep-learning methods, with higher classification accuracy. The reported results showed that the proposed approach could be used as a diagnostic tool to help the radiologist with highlighting suspicious brain regions. However, further manual inspection is needed to confirm these cases.

This method is suitable for locating and detecting tumours easily. In the future, we are planning to use other CNN models such as Squeeze-Net combined with the SVM technique and fine-tuning technique to classify tumour brain type from MRI brain images. 

## Figures and Tables

**Figure 1 entropy-24-00799-f001:**
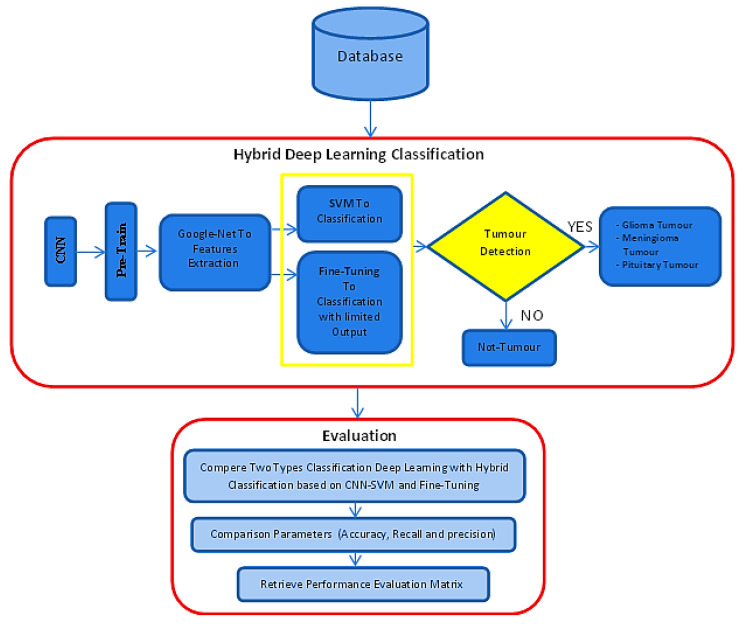
The proposed methodology.

**Figure 2 entropy-24-00799-f002:**
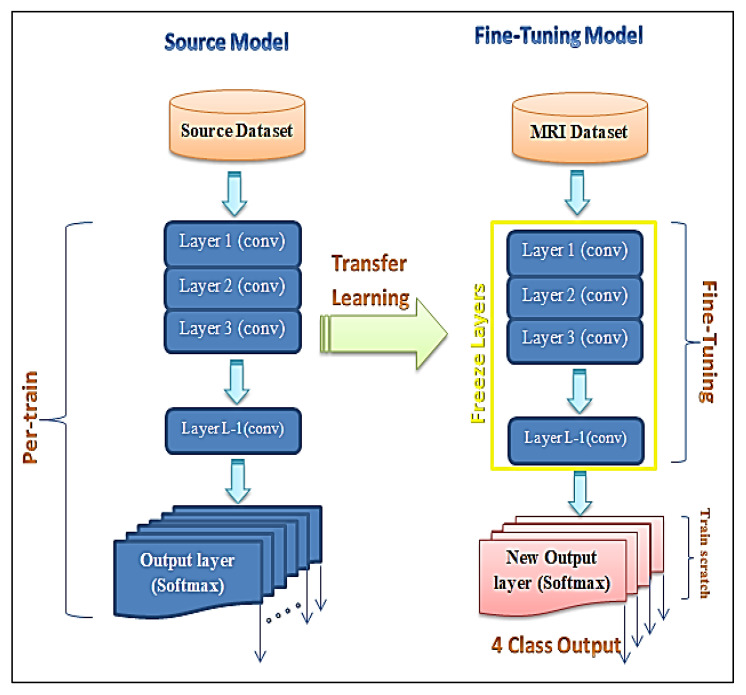
Finely tuned deep neural network model.

**Figure 3 entropy-24-00799-f003:**
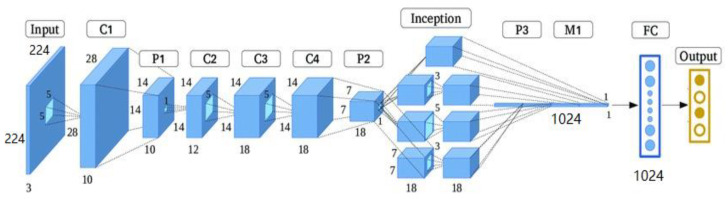
Simplified architecture of Google-Net.

**Figure 4 entropy-24-00799-f004:**
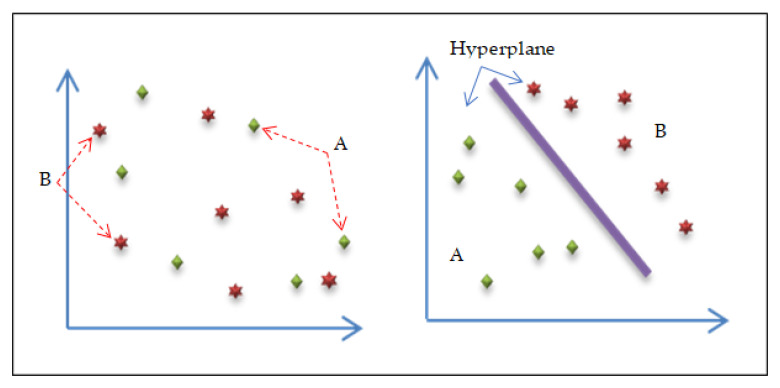
Linear SVM with the decision boundary.

**Figure 5 entropy-24-00799-f005:**
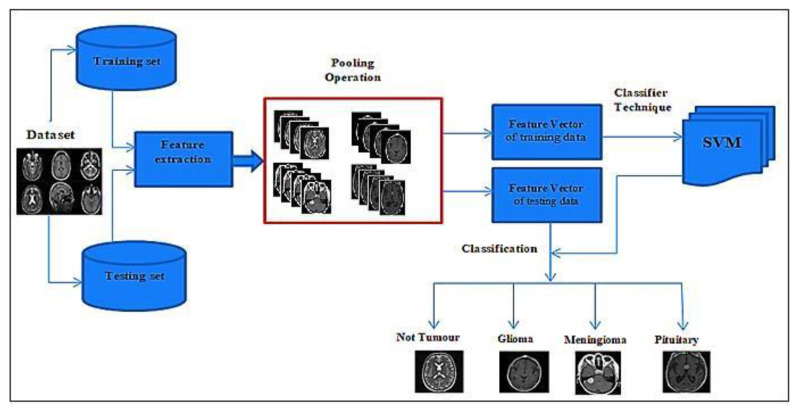
Hybrid deep-learning model (GN-SVM).

**Figure 6 entropy-24-00799-f006:**
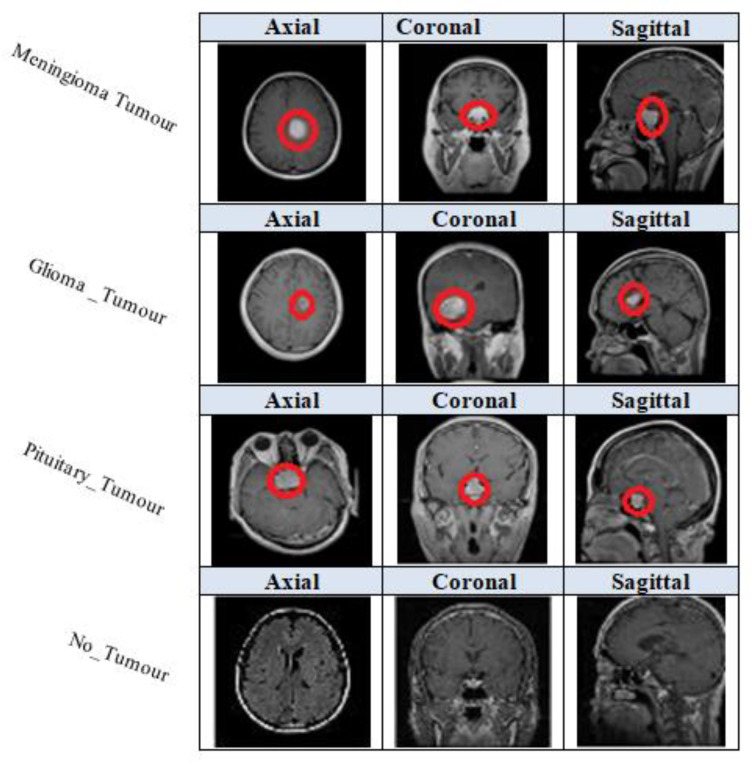
The kinds of brain tumours (shown in red circle) from three planes.

**Figure 7 entropy-24-00799-f007:**
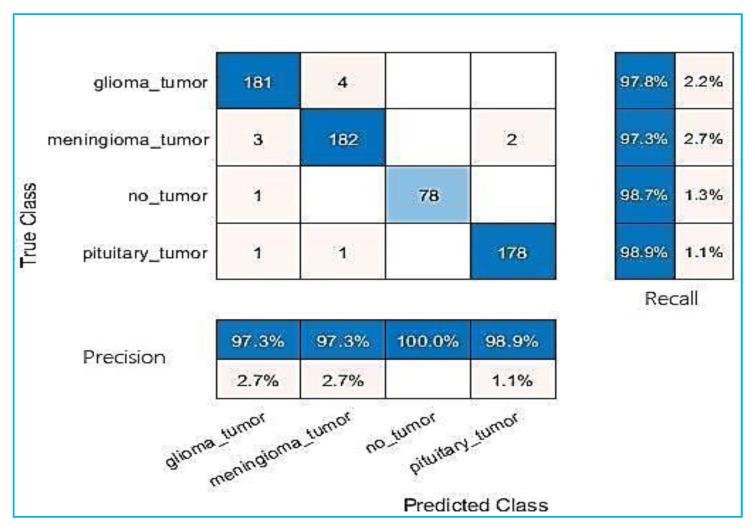
Results of Google-Net technique with SVM technique.

**Figure 8 entropy-24-00799-f008:**
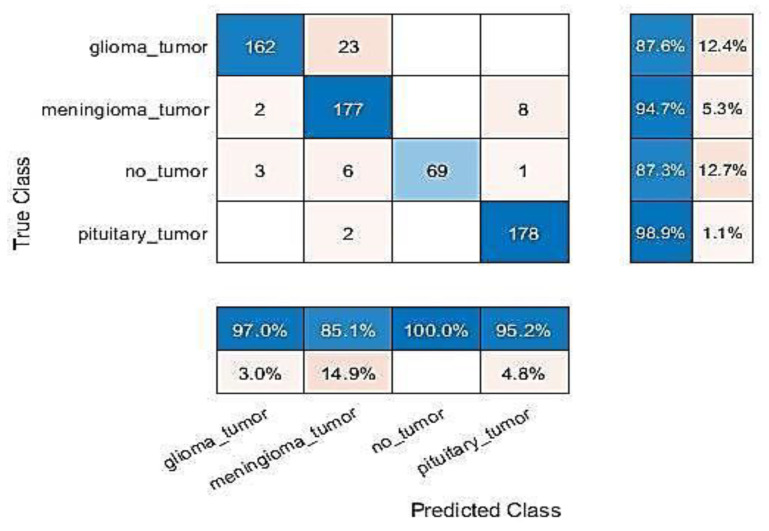
Results of finely tuned Google-Net.

**Figure 9 entropy-24-00799-f009:**
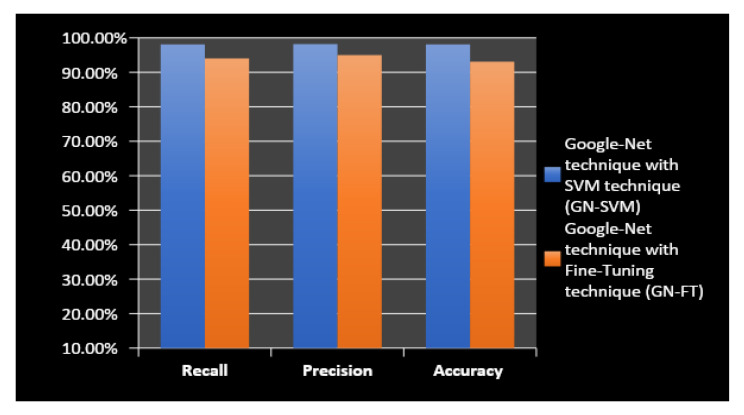
The performance of the proposed methods, GN-SVM and GN-FT.

**Figure 10 entropy-24-00799-f010:**
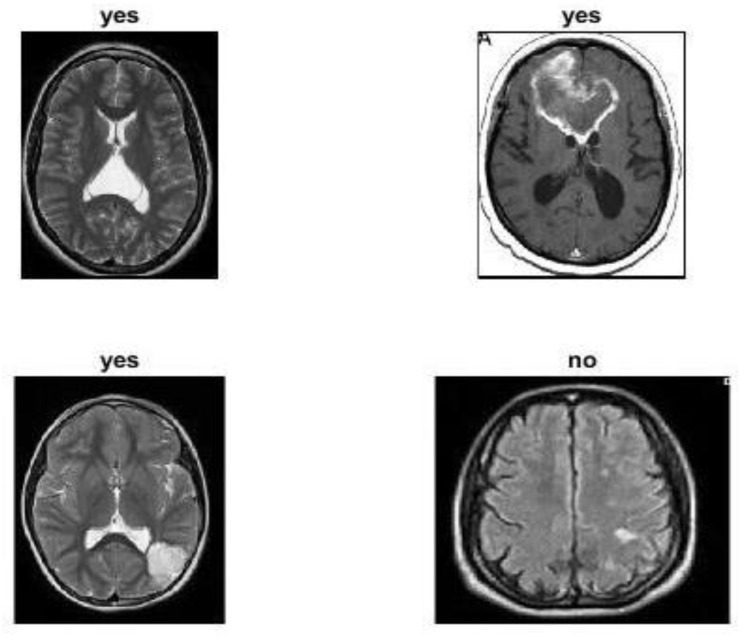
Sample images from the MRI brain dataset are publicly available.

**Table 1 entropy-24-00799-t001:** List of abbreviations.

SVM	Support Vector Machine
CNN	Convolutional Neural Network
MRI	Magnetic Resonance Images
GN-SVM	Google-Net with SVM technique
GN-FT	Google-Net with Fine-Tuning technique
WHO	World Health Organization
ML	Machine learning
DL	Deep learning
CT	Computed tomography
SPECT	Photon Emission Computer Tomography
PET	Positron Emission Tomography
ACO	Ant colony optimization

**Table 2 entropy-24-00799-t002:** Hardware and software specifications.

Item	Setting
CPU	Intel Core-I5
RAM	20 GB
Hard Drive	512 GB SSD
Operating System	Windows 10
Language	MATLAB R2021b

**Table 3 entropy-24-00799-t003:** Results of the proposed approach.

Tumour Types	Google-Net Technique with SVM Technique (GN-SVM)		Google-Net Technique with Fine-Tuning Technique (GN-FT)	
	Recall	Precision	Recall	Precision
Glioma	97.8%	97.3%	97.0%	87.6%
Meningioma	97.3%	97.3%	85.1%	94.7%
Pituitary	98.9%	98.9%	100%	87.3%
Not_Tumour	98.7%	100%	95.2%	98.9%

**Table 4 entropy-24-00799-t004:** Computational testing time analysis.

	GN-SVM	GN-FT
Test Time (second per image)	0.097	0.098

**Table 5 entropy-24-00799-t005:** Comparison with the literature.

Ref	Proposed Method	Accuracy
[27]	GLCM + SVM + BWT	96.5%
[28]	SVM + ROI + (RBF) + Linear and Cubic	97.1%
[29]	GLCM + k-NN + Fusion Operator	90.9%
[31]	GLCM + K-mean + k-NN	85.0%
[35]	Alex-Net CNN	91.2%
[36]	VGG-19 CNN	87.4% 90.7%
[37]	NS-CNN + SVM	95.6%
This paper	Google-Net + SVM	98.1%
	Google-Net + Fine-Tuning	93.1%

**Table 6 entropy-24-00799-t006:** Performance results using a public dataset.

Ref	Proposed Method	Accuracy
[34]	GLCM + k-NN + Fusion Operator	90.91%
[36]	GLCM + K-mean + k-NN	85%
[40]	Alex-Net CNN	91.16%
Proposal method	Google-Net + SVM	94.12%
	Google-Net + Fine-Tuning	90.6%

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
