# Peer review of "A Hybrid Deep Learning Model for Brain Tumour Classification"

_entropy, 2022, doi:10.3390/e24060799_

Round 1

Reviewer 1 Report

Figure 1 is very confusing. The orientation of texts is mixed. Top to bottom, or left to right data flow is preferred. The resolution of the image is also low, the image is blocky (especially the curves).

The aspect ratio of Figure 3 is not correct. Texts are stretched vertically.

Many abbreviations are used incorrectly (e.g. MRI is correctly Magnetic Resonance Imaging, PET is Positron Emission Tomography, WHO appears correctly in one place, but incorrectly in some other place).

Running time of the trained AI is measured, but the training time is not.

More technical details are required about the programming environment, programming language etc.

As the number of wrongly diagnosed images is quite low, a manual inspection of these images would be useful to make further improvements.

Author Response

Dear Reviewer , 

I would like to thank you for your valuable comments to improve the paper. All the comments has been considered as below :

Reviewer#1, Concern # 1: Figure 1 is very confusing. The orientation of texts is mixed. Top to bottom or left to right data flow is preferred. The resolution of the image is also low, the image is blocky (especially the curves).

Author response/action:  As recommended by the reviewer, we have updated Figure 1 to make it in the top to bottom data flow with a high-resolution image, as can be seen in the revised manuscript, Page 5, Figure 1.

Reviewer#1, Concern # 2: . The aspect ratio of Figure 3 is not correct. Texts are stretched vertically.

Author response/action:   We would like to thank the reviewer for his point; accordingly, the aspect ratio of Figure 3has been corrected as shown in the revised manuscript, Page 8, Figure 3.   

Reviewer#1, Concern # 3: Many abbreviations are used incorrectly (e.g. MRI is correctly Magnetic Resonance Imaging, PET is Positron Emission Tomography, WHO appears correctly in one place, but incorrectly in some other place).

Author response/action:  All abbreviations were checked and corrected, as can be seen in the revised manuscript.  

Reviewer#1, Concern # 4: Running time of the trained AI is measured, but the training time is not.

Author response/action:  It should be noted that the training time to build the model was not computed in this study because it is needed only one time. However, testing time is important, which reflects the complexity of the model, and it has been measured as given in Table 4. We have highlighted this point in the revised manuscript on Page 13.

Reviewer#1, Concern # 5: More technical details are required about the programming environment, programming language etc.

Author response/action:  both the hardware and software environment were included. This can be seen in the revised manuscript, Page 11, Table 2.

Reviewer#1, Concern # 6: As the number of wrongly diagnosed images is quite low, a manual inspection of these images would be useful to make further improvements.

Author response/action:  It should be noted the proposed hybrid model in this study will work as a tool that assists the radiologist in the diagnosis of brain tumors. However, the final decision should be taken by the radiologist. We have added a new statement in the revised manuscript that highlights this point, as can be seen in the conclusion of the revised manuscript, Page 14 

Reviewer 2 Report

I think that this paper's issue is clear and this paper is well organized. Numerical examples show a clear better performance. A little concern on this paper is the originality of the proposed methods. If this originality is theoretically explained, the paper's issue will be stronger.

Author Response

Dear Reviewer, 

I would like to thank you for your valuable comments to improve the paper.  It is worth mentioning that the main novelty of this work is related to the integration of deep learning models of pre-trained CNN as feature extractors with linear SVM. The theoretical background of both pre-trained CNN and SVM remain the same, and these models have been illustrated graphically as given in Figures 2,3 and 4. The proposed framework of this study is given in Figure, and the main contribution of this work was illustrated in Page 3, as can be found in the revised manuscript.  

This manuscript is a resubmission of an earlier submission. The following is a list of the peer review reports and author responses from that submission.

Round 1

Reviewer 1 Report

This manuscript “A HYBRID DEEP LEARNING MODEL FOR BRAIN TUMOUR CLASSIFICATION” proposes a deep learning (DL) solution to a 4-class classification problem, three brain tumour types and no-tumour.

The manuscript is very difficult to read and contains numerous grammatical, semantic and even several mishaps. Several places the brackets are used in unusual ways. There is leftover text, e.g. “This section may be divided by subheadings. It should provide a concise and precise description of the experimental results, their interpretation, as well as the experimental conclusions that can be drawn.”, which is either author’s notes or something from a template. Also the same method, e.g. GoogLeNet, is referred to differently through the manuscript. This makes the manuscript quite difficult to read. I do understand that the authors might not be native English speakers but many of the issues can be easily solved with careful proofreading. 

On the content side, it is impossible to know what was exactly done. The description of the data, how it was split into train and test sets (stratified?), whether all the MRI modalities were used simultaneously etc is missing. If all modalities were used simultaneously, then it is hard to believe that the some default feature extractor, e.g. GoogLeNet used here, would be able to extract informative features from them to achieve such a high accuracy. Such considerations should be discussed in detail. In fact almost any kind of discussion is missing from the manuscript.

Did the authors repeat the splits to avoid (un)lucky chance of getting the particular results reported? 

Also as the images come from a limited number of individuals (233) it is important that the train test splits are done carefully to avoid unintentional data leakage. For example T1 image and FLAIR image of the same individual would contain very similar information, so allowing these images to split between train and test sets would give highly optimistic results.

The results section seems to be written in quite some hurry. The figures and tables there contain almost the same information. How were the competing methods implemented? Or the authors took the numbers from previous publications? If the latter then it must be clarified as the splits might be different together with other setup like modalities, subject separation (as described above).

The figures are almost unreadable (especially the GoogLeNet) and many do not convey much information. There are typos and unclarity that can be easily improved by proofreading and actually trying to convey the message authors want to present (e.g. readers can easily look up GoogLeNet architecture which is impossible to see here). Also Fig 3 is not very helpful as readers can be easily referred to another paper or a book instead of providing a figure with insufficient description. The figure captions are not very helpful.

Quite some space is devoted to describing previous work, which can be better organised if the authors provide a logical link between the studies and what the methods mean, how they are different from others and how they compare with other methods. Currently this section is just a list of facts which does not provide much information.

Authors should refrain from using unconventional terminology such as “mission” instead of “task”.

Reviewer 2 Report

This paper presents a hybrid CNN-based framework for detecting and classifying brain tumors in MRI images. The accuracy of the model is indeed high, but the overall innovation is less, and there are already many papers similar to the work in this paper. The following points were formed after careful review:

  1. For computer or automation journals, this paper has too few innovative points, and is only a splicing of two existing models. There are already many articles with similar work, and it is recommended to submit to medical imaging journals.
  2. The brain tumor dataset used in this paper is from the brain tumor MRI image dataset of Southern Medical University, but there is a clear bias in this dataset according to a recent related study and it is said that this dataset is best done for lesion segmentation type problems. Therefore, in this paper, it is desirable for the authors to validate the proposed model using several different datasets.
  3. In the model structure, the GoogLeNet feature output part does not describe clearly what kind of image features are extracted, and the impact on the subsequent SVM classification is not explained, so the overall working process of the model seems too vague.